# Dynamic and Static Splinting for Treatment of Developmental Dysplasia of the Hip: A Systematic Review

**DOI:** 10.3390/children8020104

**Published:** 2021-02-04

**Authors:** Vito Pavone, Claudia de Cristo, Andrea Vescio, Ludovico Lucenti, Marco Sapienza, Giuseppe Sessa, Piero Pavone, Gianluca Testa

**Affiliations:** 1Department of General Surgery and Medical Surgical Specialties, Section of Orthopaedics and Traumatology, University Hospital Policlinico-Vittorio Emanuele, University of Catania, 95123 Catania, Italy; decristo.claudia@gmail.com (C.d.C.); andreavescio88@gmail.com (A.V.); ludovico.lucenti@gmail.com (L.L.); marcosapienza09@yahoo.it (M.S.); giusessa@unict.it (G.S.); gianpavel@hotmail.com (G.T.); 2Department of Clinical and Experimental Medicine, Section of Pediatrics and Child Neuropsychiatry, University of Catania, 95123 Catania, Italy; ppavone@unict.it

**Keywords:** developmental dysplasia of the hip, DDH, treatment, conservative, bracing, dynamic splint, static splint

## Abstract

Background: Developmental dysplasia of the hip (DDH) is one of the most common pediatric conditions. The current gold-standard treatment for children under six months of age with a reducible hip is bracing, but the orthopedic literature features several splint options, and each one has many advantages and disadvantages. The aim of this review is to analyze the available literature to document the up-to-date evidence on DDH conservative treatment. Methods: A systematic review of PubMed and Science Direct databases was performed by two independent authors (C.d.C. and A.V.) using the keywords “developmental dysplasia hip”, “brace”, “harness”, “splint”, “abduction brace” to evaluate studies of any level of evidence that reported clinical or preclinical results and dealt with conservative DDH treatment. The result of every stage was reviewed and approved by the senior investigators (V.P. and G.T.). Results: A total of 1411 articles were found. After the exclusion of duplicates, 367 articles were selected. At the end of the first screening, following the previously described selection criteria, we selected 29 articles eligible for full text reading. The included articles mainly focus on the Pavlik harness, Frejka, and Tubingen among the dynamic splint applications as well as the rhino-style brace, Ilfeld and generic abduction brace among the static splint applications. The main findings of the included articles were summarized. Conclusions: Dynamic splinting for DDH represents a valid therapeutic option in cases of instability and dislocation, especially if applied within 4–5 months of life. Dynamic splinting has a low contraindication. Static bracing is an effective option too, but only for stable hips or residual acetabular dysplasia.

## 1. Introduction

Developmental dysplasia of the hip (DDH) is a common pediatric condition that has a variable incidence due to the genetic predisposition and cultural practices of different ethnicities [1]. DDH consists of a spectrum of abnormalities that range from delayed physiological development of the hip, mild capsular laxity, to acetabular deficiency, subluxation, and dislocation of the hip.

The etiology of DDH is multifactorial, involving both genetic and intrauterine factors. The gold standard for imaging infant hips is ultrasonography (US). The Graf classification system is the most adopted system for classifying infant hips basing on US images [2,3]. Radiographs may be useful starting at 4–6 months of age, but it is more suitable after femoral head ossification, which occurs by six months of age in 80% of infants [4].

The treatment of DDH has undergone significant evolution in the last few decades, depending on the patient’s age and the severity of the condition. Due to abduction and flexing of the hips while they are worn, splints and braces are applicated in different diseases [5], and are actually considered the gold standard for DDH-affected children under 6 months of age with a reducible hip. The dynamic splint promotes a “dynamic reduction”: the child can move his/her legs within the range permitted by the splint, maintaining the hips in flexion and abduction while restricting extension and adduction. The Pavlik harness is the most popular dynamic splint. Other dynamic splints used for treating DDH are the Tubingen splint, Frejka pillow, Von Rosen splint, Aberdeen splint, Coxaflex and Teufel brace [6,7,8,9,10,11,12]. The static splints promote a “rigid reduction”. They consist of a metallic or hard plastic support that keeps the legs of the child in a fixed position of abduction and flexion, without the possibility of hip motion. They seem to have a higher rate of complications compared with avascular necrosis (AVN); therefore, they are less commonly used [13]. The most common static harnesses are the rhino brace, Denis Browne bar, Milgram brace and the Ilfeld harness. Generally, a dynamic splint is indicated for a reducible hip in patients that are not yet able to stand. The most accepted indication is an unstable hip that can be centered without the need for a spica cast [14]. On the other hand, static splints are an effective alternative to the dynamic splints for children more than 6–9 months of age who require continued abduction positioning because of acetabular dysplasia and/or subluxation [15].

All of the treatments with splints have risks of avascular necrosis (AVN) and femoral nerve palsy [16,17]. Higher rates of AVN are reported after unsuccessful hip reduction, presentation beyond 3 months of age, fixed dislocation and bilateral hip involvement [18,19,20,21]. The “Pavlik harness disease” is a complication following an inappropriate continuation of the harness with a dislocated hip. Femoral nerve palsy occurs on the involved side in 2.5% of patients treated with a dynamic splint, usually in the first week of treatment, and resolves within two weeks. This complication was shown to be strongly predictive of treatment failure [16].

There are few data in the literature regarding the differences between dynamic and static splints and the different varieties of each type. The aim of this study is to clarify the differences between the success, failure and complications rates of several braces available for the treatment of DDH.

## 2. Materials and Methods

### 2.1. Study Selection

According to the guidelines of the Preferred Reporting Items for Systematic Reviews and Meta-Analyses (PRISMA) [22], a systematic review of PubMed and Science Direct databases was performed by two independent authors (C.d.C. and A.V.) using the keywords “developmental dysplasia hip”, “brace”, “harness”, “splint”, and “abduction brace”. Previous keywords or MeSH terms were combined in order to achieve the maximum research efficacy.

From each included article, a standard data entry form was utilized to extract the number of patients, number of hips treated, affected side, sex, age of patient at start of treatment, type of DDH, duration of splinting, number of successes and failures, success and failure according to the grade of DDH, complication rate and complication type, follow-up and period of the study.

The risk of bias assessment was performed by two independent reviewers (C.d.C. and A.V.) using the Dutch checklist form for prognosis recommended by the Cochrane Collaboration. The checklist was applied with modifications to the items that were relevant to the current study’s objectives [22]. Conflicts were resolved by consultation with a senior surgeon (V.P.). Table 1 represents the risk of bias summary including the checklist items. Items could be scored as ‘low risk’ (+), ‘high risk’ (−), or ‘unclear’ (?). The forms were then compared and discussed to achieve a final consensus (Table 1).

### 2.2. Inclusion and Exclusion Criteria

Eligible studies for the present systematic review included DDH treatment and splintage. The initial titles and abstracts screening was performed using the following inclusion criteria: treatment consisted of hip bracing without operative treatment or cast application in children aged under one-year, with a minimum average of four-months follow-up. The exclusion criteria were groups of patients with secondary hip dysplasia, including syndromic and teratogenic DDH, hip surgery treatment and casting. We also excluded all remaining duplicates, articles dealing with other topics, those with poor scientific methodology or those without an accessible abstract. Reference lists were also hand-searched for further relevant studies. Abstracts, case reports, conference presentations, editorials and expert opinions were excluded.

Two classification systems were considered: the Graf system based on US and a clinical classification according to hip stability, identifying a hip as stable, dislocatable (Barlow positive), reducible (femoral head dislocated but reducible by the Ortolani maneuver) and irreducible. For those studies that did not differentiate the subtypes of Type II into groups A, B, C and D, we considered all type II hips as type IIB in order to standardize the sample. Similar to what Grill did for his study [28], for evaluation reasons, we merged hips of grade Tönnis 1 with the type Graf IIb, grade Tönnis 2 with type Graf III, and Tönnis 3 and 4 with type Graf IV. This correlation made it possible to evaluate the material in one block. For the articles where the total number of hips was not specified, we considered the number of children to represent the number of hips.

### 2.3. Definition of Outcomes

We considered success to be treatment resulting in the regression of the dysplasia with recovery of the hip. In the case of hips that were irreducible, unstable at rest or on stress exam, not improved at follow-up, or when the infants underwent splinting or bracing change, spica cast application or surgical management, the treatment was assessed as “unsuccessful”. On the other hand, progression of dysplasia within the first 4–8 weeks and the need for further and more invasive treatments, including casting in general anesthesia, were considered as failure. We considered only major complications that included AVN and femoral nerve palsy or other nerve palsies.

### 2.4. Statistical Analysis

Review Manager 5.4.1 (Review Manager (RevMan), The Cochrane Collaboration, 2020) was used to perform the meta-analysis of the selected articles that applied comparable records descriptions and had similar study cohorts. Odd ratios were combined, using the generic inverse variance. The fixed-effects model was used for all meta-analysis.

## 3. Results

A total of *n* = 1411 articles were found, including three articles added after the reference list analysis. After the exclusion of duplicates, *n* = 367 articles were selected. At the end of the first screening, following the previously described selection criteria, we selected *n* = 29 articles for full-text reading. Ultimately, after reading the full texts and checking the reference lists, we selected *n* = 19 articles following the previously written criteria. In Table 2 the main findings are reported according to principal author and brace/splint.

A PRISMA [21] flowchart of the method of selection and screening is provided (Figure 1).

In these 19 studies, a total of 5100 patients and 6755 hips were identified. Overall, a success with splintage was observed in 6272 hips with a total success rate of 93%. Fourteen studies concerned dynamic splints (74%) (dynamic splint group), and five studies were about static braces (26%) (static brace group).

The included articles [9,10,11,12,13,14,15,16,17,18,19,20,21,22,23,24,25,26,27,28,29,30,31,32,33,34,35,36,37] mainly focus on Pavlik harness, Frejka, and Tubingen among the dynamic splints, while the rhino-style brace, Ilfeld and generic abduction brace were considered among the static splints. The main findings of the included articles are summarized in Table 3 and Table 4.

Overall, a success with splintage was observed in 5515 hips with a total success rate of 93%. Considering only the dynamic splint group, treatment success was reached in 5287 hips with a rate of 91.3%, while a failure occurred in 427 hips with a failure rate of 8.7% (Table 4). The static brace group had a success rate of 93.0% with 212 hips healed, and a failure rate of 7.0% with 16 hips failed (Figure 2). According to the success rate, no statistically significant difference was noted between dynamic and static splinting/bracing (*p* = 0.63).

Overall, the average follow-up of the studies was 36.4 months (range 2–168). The average age of patients at the start of the treatment was 6.8 weeks for the dynamic splint group (range 0.1–40) and 7.5 weeks for the patients treated with the static brace (range 1–19 weeks). The average full-time duration of splinting was 16.4 weeks (range 5–25.2) for dynamic splinting and 8.9 weeks (range 0.5–32) for the static group. Only four studies (21%) reported part-time wearing of the brace (only night-time wearing) for a mean of 10.4 weeks (range 6–16.4). Globally, a major complication occurred in 141 of the 6755 hips treated, with a rate of 2%. 136 were AVN (2%), four femoral nerve palsy (0.05%) and one growth arrest line (0.01%) over the whole sample. Among the dynamic splint group, the rate of complication was 2.1% with a total of 138 cases. The AVN rate was 2% with 134 cases. Regarding the static brace group, a complication was observed in three cases (1.3%), and two of them were AVN (AVN rate 0.8%) (Table 5).

### 3.1. Abduction Brace

160 hips treated with an abduction brace were described in three different studies: Eberle et al. [27] reported that 137 of 139 hips were successfully treated, Hedequist et al. [29] found 14% (two cases) with complications including one AVN and Ibrahim et al. [30] reported only one AVN case.

### 3.2. Aberdeen Splint

Only one study reported DDH treatment with an Aberdeen splint [37]: Williams et al. in 1999 reported a 120-patient sample with a 98.3% success rate and a 2% rate of complications.

### 3.3. Coxaflex splint

Azzoni et al. (2011) [12], in a comparison study, reported 58 out of 59 successfully treated hips (98.3%), and no complications were observed in the sample.

### 3.4. Craig Splint

Wilkinson et al. in 2002 [36] reported the only Craig splint study that was included in the systematic review. They investigated 22 patients (28 DDH hips) and reported a 85.7% success rate with no complications.

### 3.5. Frejka pillow

A success rate between 89% [26] and 97.2% [34] and a complication rate between 0.9% and 12%, AVN in every case, were reported in four studies [11,24,26,34] investigating DDH patient treatment with the Frejka pillow.

### 3.6. Ilfeld Splint

Sankar et al. (2015) [33] evaluated 28 hips, reporting 82.1% successful treatments and no complications.

### 3.7. Pavlik harness

The Pavlik harness is the most represented splint with seven studies [18,24,25,26,28,31,36] included. These comprised a sample of 4779 hips with a success rate of 91.6%, a complication rate of 2.3% and a total of 105 AVN cases. Grill et al. [28] in 1988 described 3611 hips treated with the Pavlik harness and reported only a 7.9% rate of failure. Novais et al. [31] is the only selected study that reported four femoral nerve paralysis cases.

### 3.8. Rhino-Style Splint

Wahlen et al. [35], the only rhino-style splint study included in the systematic review, investigated 40 hips (33 patients) and reported a success rate of 87.5% and no complications.

### 3.9. Teuffel Splint

Azzoni et al. [12], in a comparison study, reported that all 59 hip treatments were successful and that no complications were observed in the sample.

### 3.10. Tubingen Splint

Three studies [9,23,32] investigated DDH patients treated with Tubigen splint, the largest sample after Pavlik brace (713 hips), and reported a success rate of 97.5% and a complication rate of 0.4%.

### 3.11. Von Rosen Splint

A 100% success rate is described for Von Rosen in 333 patients. Both Hilderaker et al. [11] and Wilkinson et al. [36] did not report complications.

## 4. Discussion

This analysis of DDH 6755 cases from the literature is one of the first systematic reviews to compare the outcomes and complication rates of dynamic and static hip splints. A variation of treatment timing, modalities and splints were described among the orthopedics. Dynamic splints are currently the preferred choice. According to a recent study, the type of brace that is most widely used is the Pavlik harness, accounting for 70–90% of Pediatric Orthopaedic Society of North America (POSNA) and European Paediatric Orthopaedic Society (EPOS) members, while rigid braces are chosen only in 20%, the Frejka pillow in 13% and the Von Rosen splint in less than 10% [6].

The success rate of the Pavlik harness was found to be 91.1%. Few studies have reported the efficacy of different splints: some compared the use of the Frejka pillow with the Von Rosen splint, and others described the Craig and the Von Rosen splints to be slightly superior to the Pavlik harness [27,38,39]. Surely, the Pavlik harness remains the most preferred treatment for the majority of children younger than 6 months, as it is the most thoroughly described and analyzed and found to be safe and highly effective with large samples [10,23]. The most satisfying outcomes were described with the use of Tubingen (97.5%), Von Rosen (100%), Aberdeen (98.3%%), Coxaflex (98.3%%) and Teufel (100%) splints, but the small sample of the latter three braces does not aid comparison. An increase in successful outcomes of the static braces was observed over the last few years, but only five studies for a total of 228 hips were included in the study. This may be attributable to improvements in the achievement of custom-made braces over time and to more numerous cases of low- and mid-grade of dysplasia being encountered. A good advantage is that the design avoids any maladjustment for the user, unlike Pavlik’s harness, so that the brace is easily applied without any risk of improper positioning; moreover, static braces should be also used in children older than 6 months [35].

We found a success rate of the static brace group of 93.0% and a failure rate of 7.0%. Even if the number of total cases included is only 212 hips, the static brace seems to be effective. Several authors reported series of patients in whom static bracing successfully stabilized persistent posterior dislocations following Pavlik harness failure [27,29,33,40], and in three of these studies [27,29,30], the static brace was applied after Pavlik treatment failure, causing a decrease in the percentage of success. In fact, there were a total of 49 hips with a success rate of 69.3%, but with a much higher rate of failure concerning irreducible and dislocatable but reducible hips.

Static bracing has been previously described as a viable option for treating DDH after a Pavlik failure and has the obvious advantages of avoiding both general anesthesia and a spica cast, which expose the child to several risks [41]. Eighty-two percent of hips that fail using the Pavlik harness respond successfully to rigid hip abduction bracing, but if rigid bracing cannot reduce the hip at the beginning of treatment, it should be avoided [27,31]. On the other hand, Ibrahim et al. [30] demonstrated opposite results, with a 100% of failure over seven hips, finding no advantage of static splinting but only an unnecessary delay of the time to closed reduction. However, it should be noticed that three patients had irreducible hips. Observing these series, even after an initial failure of dynamic splint treatment, an attempt of treatment with a static brace should be done for 3–4 weeks, but in our opinion, the condition is that the hip must be well located in the acetabulum. However, this switch is generally preferred by American surgeons and poorly performed by European physicians [6,29]. The mechanism by which a static splint may succeed where the Pavlik harness failed is unclear. The most probable reason may be related to inferior dislocation aggravated by flexion. A rigid or semirigid brace generally holds hips in less flexion than a standard Pavlik harness fitting and may be useful for certain dislocations that are predominantly inferior. The rigidity of the device may provide more structure for certain hips that remain excessively lax even within a Pavlik harness. It may also be that rigid orthoses make it easier to apply for certain families [29,42].

Even though dynamic splints observe the safe position suggested by Ramsey [43], spontaneous AVN remans the main complication in dysplastic hip treatment and must be considered as iatrogenic secondary to splintage [44]. Several studies report the AVN rate as negligible when the splints straps are properly adjusted [18,43,45]. The complication rate of the Pavlik harness in these series is 2.3% with an AVN rate of 2.2%. The Frejka pillow has the highest AVN rate, with 5.6% of 436 hips, due to the extreme abduction of this splint. Even if this review reports high rates of success and few complications for the Von Rosen splint, one should be considerate of the small sample. Also, the degrees of abduction seem to be too extreme for a safe treatment. Even if several studies found an increased risk of AVN with more rigid braces, we found an AVN rate of 0.8%, and if we do not consider the irreducible hips, the value decrease further. Static splints should be used only if the epiphysis is well positioned in the acetabular base, because this type of brace can often fail to center the hip in the acetabulum, and especially because a not-well-centered epiphysis will be strongly stressed against the superior roof of the acetabulum by a rigid splint, damaging the cartilaginous component and delaying healing, with the possibility of worsening the dislocation [13]. We did not find a correlation between AVN rate and duration of treatment. The treatment of residual acetabular dysplasia in children older than six months remains challenging. Spontaneous resolution of this residual dysplasia without intervention is unlikely in children over six months of age [46]. Some studies support the use of static splints to treat residual acetabular dysplasia in older infants when they have outgrown the Pavlik harness, improving the acetabular index, but the data are still limited [39,47,48]. However, a part-time rigid abduction brace is often used at many centers to produce some improvement in the acetabular index without a major impact on the child’s activity, but the optimal duration of the brace is still unclear [39].

An important limit is the improper positioning of the brace, which is often an iatrogenic cause of damage. Insufficient hip flexion is one of the most common pitfalls that can lead to an insufficient reduction. In dynamic braces/splints, another risk is that the distal components can be positioned too distally at the level of the knee. The incorrect position of the distal part of brace could cause a limited hip reductive effect in the acetabulum due to a lever mechanism [13]. A cause of failure is also parents’ low cooperation with the use of the brace. Parents play a key role in the effective use of the splint, and they must be educated about the proper use of the harness to increase the chance of success [14]. Regarding the approach of application of the splint, the recent survey of EPOS and POSNA members found that around 20% of surgeons allowed their patients to always wear clothes underneath their brace, 15% never allowed clothes, 20% only allowed clothes once the hip was clinically stable, and about 22% only allowed underwear [6].

This study has several limitations, including the sample heterogeneity for number, age of population and grade of dysplasia and the bias risk related to no common definition of successful outcome. Larger samples, comparative studies and defined quality standards are needed, especially for the static braces. The classification system of the grade of DDH is not homogeneous in all the articles included; therefore, there exists bias in the analysis. Several studies have short-term follow-ups that cannot accurately verify the success of the treatment and the presence of a delayed AVN.

## 5. Conclusions

Dynamic splinting for DDH represents a valid therapeutic option in cases of instability and dislocation, especially if applied within 4–5 months of life. Dynamic splinting has a low contraindication and is very well tolerated. The Pavlik harness is still the most used brace, but the Tubingen splint showed better outcomes with major tolerance and compliance. The limits concern the accurate indications and timing of initiation. The static brace is an effective option too, but only for stable hips: it is imperative that the femoral head be well centered in the acetabular base for a safe treatment. Static braces can be also useful in cases of residual acetabular dysplasia.

## Figures and Tables

**Figure 1 children-08-00104-f001:**
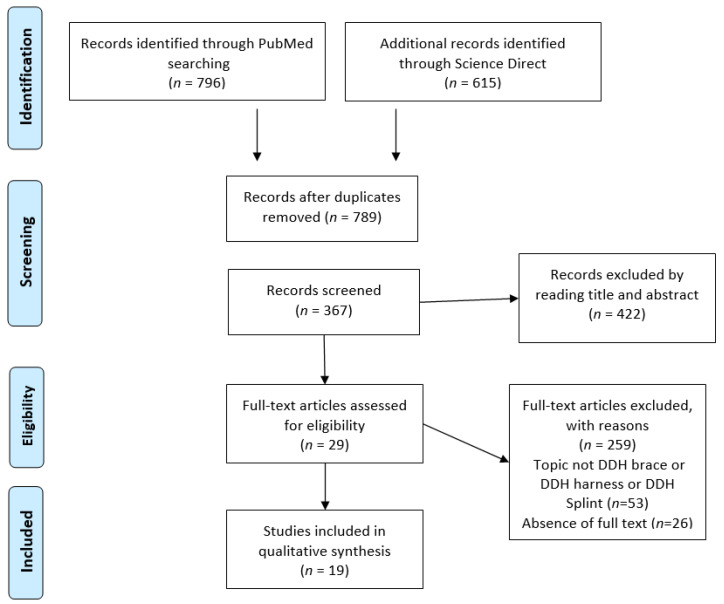
PRISMA (Preferred Reporting Items for Systematic Reviews and Meta-Analysis) flowchart of the systematic literature review.

**Figure 2 children-08-00104-f002:**
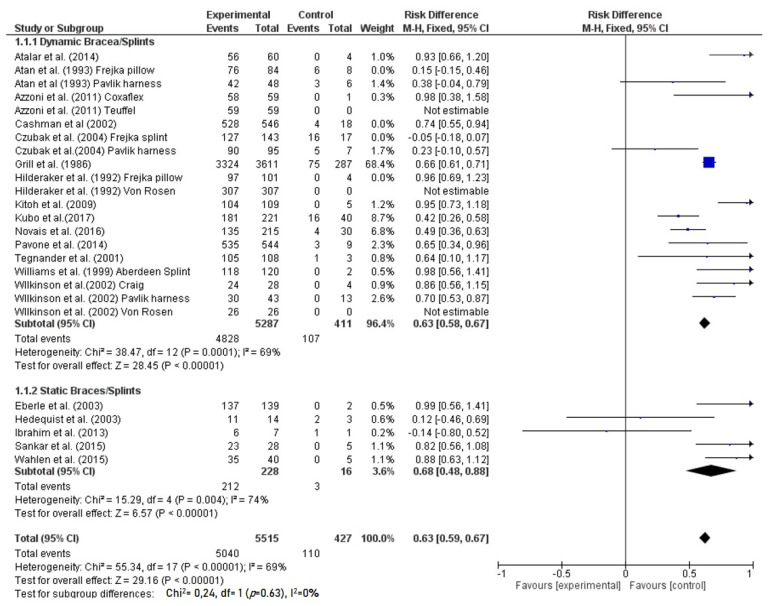
Forest plot of comparison: dynamic vs. static bracing/splinting. M-H = Mantel-Haenszel method; CI = confidence interval; arrow = overall effect, square = point estimate and confidence intervals of study; diamond = point estimate and confidence intervals for type of brace/splint.

**Table 1 children-08-00104-t001:** Risk of bias of the included studies.

Ref	Author	No Participant Selection Took Place	Groups Are Comparable Regarding Age	Validated Measuring System Used	Independent (Blind) Determination of Outcomes	Clear Description of Groups Available
[23]	Atalar H. et al. (2014)	+	+	+	?	+
[24]	Atan D et al. (1993)	+	?	−	?	+
[12]	Azzoni et al. (2011)	+	?	+	+	−
[25]	Cashman et al. (2002)	+	?	+	?	+
[26]	Czubak et al. (2004)	?	+	+	?	−
[27]	Eberle et al. 2003	+	?	−	?	+
[28]	Grill et al. (1988)	−	+	+	?	+
[29]	Hedequist et al. (2003)	+	−	+	?	+
[11]	Hilderaker et al. (1992)	+	+	+	?	+
[30]	Ibrahim et al. (2013)	+	?	+	?	+
[9]	Kubo et al. (2018)	+	+	−	?	+
[18]	Kitoh et al. (2009)	+	?	+	+	+
[31]	Novais et al. (2016)	+	?	+	?	+
[32]	Pavone et al. (2015)	+	?	+	?	+
[33]	Sankar et al. (2015)	+	+	+	?	+
[34]	Tegnander et al. (2001)	+	?	−	?	+
[35]	Wahlen et al. (2015)	+	+	+	?	+
[36]	Wilkinson et al. (2002)	−	?	+	?	+
[37]	Williams et al. (1999)	+	?	+	?	+

+: low risk; −: high risk; ?: unclear.

**Table 2 children-08-00104-t002:** Main findings of the included studies (No. = number; f = female; m = male; ? = unknown; AVN = avascular necrosis).

Ref	Author	Brace	No. of Patients (Females, Males)	No. of Hips	Mean Age of the Brace at Treatment Onset (Weeks)	Follow Up (Months)	Success Event (%)	Complications Event (%)	AVN
[23]	Atalar et al. (2014)	Tubingen splint	49 (45 f, 4 m)	60	18	24	56 (93.3)	0 (0)	0 (0)
[24]	Atan et al. (1993)	Frejka pillow	70 (54 f, 16 m)	84	0.7	?	76 (90.5)	6 (7)	6 (7)
[24]	Atan et al. (1993)	Pavlik harness	40 (29 f, 11 m)	48	1.4	?	42 (87.5)	3 (6)	3 (6)
[12]	Azzoni et al. (2011)	Teuffel	?	59	6.1	?	59 (100)	0 (0)	0 (0)
[12]	Azzoni et al. (2011)	Coxaflex	?	59	6.1	?	58 (98.3)	0 (0)	0 (0)
[25]	Cashman et al. (2002)	Pavlik harness	332 (275 f, 57 m)	546	?	?	528 (96.7)	4 (1)	4 (1)
[26]	Czubak et al. (2004)	Pavlik harness	95 (84 f; 11 m)	?	11.5	?	90 (95)	7 (7)	7 (7)
[26]	Czubak et al. (2004)	Frejka splint	143 (129 f; 14 m)	?	11.5	?	127 (89)	17 (12)	17 (12)
[27]	Eberle et al. (2003)	Abduction brace	113	139	?	40	137 (99)	0 (0)	0 (0)
[28]	Grill et al. (1988)	Pavlik harness	2636 (2343 f; 293 m)	3611	5	53.5	3324 (92.1)	75 (2)	75 (2)
[29]	Hedequist et al. (2003)	Abduction brace	?	14	3.7	12	11 (79)	2 (14)	1 (7)
[11]	Hilderaker et al. (1992)	Frejka pillow	101	?	?	?	97 (96)	0 (0)	0 (0)
[11]	Hilderaker et al. (1992)	Von Rosen	307	?	?	?	307 (100)	0 (0)	0 (0)
[30]	Ibrahim et al. (2013)	Abduction brace	7 (7 f; 0 m)	7	10.9	33.6	6 (86)	1 (14)	1 (14)
[9]	Kubo et al. (2018)	Tubingen splint	79 (74 f; 5 m)	109	3.1	24	104 (95.4)	0 (0)	0 (0)
[18]	Kitoh et al. (2009)	Pavlik harness	210(190 f; 20 m)	221	15.6	12	181 (81.9)	16 (7)	16 (7)
[31]	Novais et al. (2016)	Pavlik harness	135 (107 f; 28 f)	215	4.3	4	185 (86.0)	4 (2)	0 (0)
[32]	Pavone et al. (2015)	Tubingen splint	351 (248 f, 103)	544	9.7	76.8	535 (98.3)	3 (0.6)	3 (0.6)
[33]	Sankar et al. (2015)	Ilfeld	19	28	4.6	12	23 (82.1)	0 (0)	0 (0)
[34]	Tegnander et al. (2001)	Frejka pillow	108	?	16	?	105 (97.2)	1 (0.9)	1 (0.9)
[35]	Wahlen et al. (2015)	Lausanne brace (rhino-style)	33	40	11	40	35 (87.5)	0 (0)	0 (0)
[36]	Wilkinson et al. (2002)	Craig	22	28	5.3	?	24 (85.7)	0 (0)	0 (0)
[36]	Wilkinson et al. (2002)	Pavlik harness	30	43	7	?	30 (69.3)	0 (0)	0 (0)
[36]	Wilkinson et al. (2002)	Von Rosen	16	26	3.7	?	26 (100)	0 (0)	0 (0)
[37]	Williams et al. (1999)	Aberdeen Splint	86	120	?	108	118 (98.3)	2 (2)	2 (2)

**Table 3 children-08-00104-t003:** Number of samples collected for each type of dynamic splint.

Type of Dynamic Splint	Number of Hips	Proportion within Dynamic Group
Pavlik	4779	73.2%
Tubingen	713	14.9%
Frejka pillow	436	9.1%
Von Rosen	333	5.1%
Aberdeen	120	1.8%
Coxaflex	59	0.9%
Teufel	59	0.9%
Craig	28	0.4%
TOTAL	6527	

**Table 4 children-08-00104-t004:** Number of sample collected for each type of static brace.

Type of Static Brace	Number of Hips	Proportion within Static Group
Abduction brace	160	70.2%
Rhino	40	17.5%
Ilfeld	28	12.3%
TOTAL	228	

**Table 5 children-08-00104-t005:** Complications and AVN.

	Dynamic Splint Group	Static Brace Group	Total
Complication (No. of hips)	138	3	141
Complication rate	2.1%	1.3%	2%
AVN (No. of hips)	134	2	136
AVN rate	2%	0.8%	2%

No. = number; AVN = avascular necrosis.

## Data Availability

No new data were created or analyzed in this study. Data sharing is not applicable to this article.

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
