# Peer review of "Dynamic and Static Splinting for Treatment of Developmental Dysplasia of the Hip: A Systematic Review"

_children, 2021, doi:10.3390/children8020104_

Round 1

Reviewer 1 Report

Review of Pavone et. al.

Abstract,

Do you need to use n= when giving the number of manuscripts.  Just say “A total of 1411 articles . . . “  

Next to last sentence, “Dynamic splinting has few contraindications 

Last sentence “The static brace is also an effective option . .

Introduction

2nd paragraph, 2nd sentence.  I would say the gold standard for imaging infant hips is ultrasonography.  Perhaps in Europe US is used on every hip, but in the US it is only for select cases.  This I would disagree that it is the gold standard for evaluating, but it is certainly the gold standard for imaging.

Methods

2.2, 1st paragraph:  What is mean by “poor scientific methodology”  Did the authors have strict criteria?  This needs to be explained.

2.3  With a minimum average follow up of only 4, I am not sure the authors can say it was truly successful treatment.  The first sentence is confusion about what is in the parentheses (surgical or nonsurgical treatment).  Please clarify.  What other major complications were considered besides AVN and femoral N palsy – please state all the complications considered.

Paragraph just below Table 4:  The reader can figure out that the failure rate was 7% by simple math – just delete that sentence.

Table 5:  Is there any way a p value for the difference between the 9.7% and 7% failure rates can be determined in a meta-analysis?  That would be interesting to know if it can be done.  Please consult with a meta-analysis statistician person. 

Discussion:  next to last paragraph – I don’t understand al what is mean by “creating with the knee flexion a lever mechanism on the proximal epiphysis”  Please explain better.  And in this paragraph, it seems the authors are going between both static and dynamic bracing, please clarify and organize – unless I am missing something.  

Discussion:  Last sentence “Further studies . . . “.  Please delete.  You have addressed the limitations fo the study.

Author Response

Abstract

Q1) Do you need to use n= when giving the number of manuscripts.  Just say “A total of 1411 articles . . . “ 

A1) Thanks for your comment. The requested modifies were made.

Q2) Next to last sentence, “Dynamic splinting has few contraindications, Last sentence “The static brace is also an effective option . .

A2) Thanks for your comment. The requested modifies were made.

Introduction

Q3) 2nd paragraph, 2nd sentence.  I would say the gold standard for imaging infant hips is ultrasonography.  Perhaps in Europe US is used on every hip, but in the US it is only for select cases.  This I would disagree that it is the gold standard for evaluating, but it is certainly the gold standard for imaging.

A3) Thanks for your comment. The requested modifies were made

Methods

Q4) 2.2, 1st paragraph:  What is mean by “poor scientific methodology”  Did the authors have strict criteria?  This needs to be explained.

A4) Thanks for your comment. Poor scientific methodology was assessed according to the Dutch checklist form as reported in the study selection method and the results were included in table 1.

Q5) 2.3  With a minimum average follow up of only 4, I am not sure the authors can say it was truly successful treatment. 

A5) Thanks for your comment. We agree with the reviewer suggestion, 4-months follow-up is a too short period to assess the treatment success. On the other hand, in our study, Novais et al. article is the only one with a 4-months follow-up and reported 4 complication out 215 patients. The manuscript was considered valid after the risk bias assessment.  For this reason, the article was included in the study.

Q6) The first sentence is confusion about what is in the parentheses (surgical or nonsurgical treatment).  Please clarify.

A6) Thanks for the comment. The requested clarification was added.

Q7) What other major complications were considered besides AVN and femoral N palsy – please state all the complications considered.

A7) Thanks for the comment. According to Dwan, Kerry et al. “Splinting for the non‐operative management of developmental dysplasia of the hip (DDH) in children under six months of age.” The Cochrane Database of Systematic Reviews vol. 2017,7, in addition to AVN and femoral N palsy, “other nerve palsies” was evaluated as “Major complication”. ”Pressure areas on skin” was evaluated as “minor complication”. The requested clarification was added in the text.

Q8) Paragraph just below Table 4:  The reader can figure out that the failure rate was 7% by simple math – just delete that sentence.

A8) Thanks for your comment. The requested modifies were made

Q9) Table 5:  Is there any way a p value for the difference between the 9.7% and 7% failure rates can be determined in a meta-analysis?  That would be interesting to know if it can be done.  Please consult with a meta-analysis statistician person.

A9) Thanks for your suggestion, we believe that the statistical analysis of our result could improve the study quality. A proper subsection was added in methods.

Q10) Discussion:  next to last paragraph – I don’t understand al what is mean by “creating with the knee flexion a lever mechanism on the proximal epiphysis”  Please explain better.  And in this paragraph, it seems the authors are going between both static and dynamic bracing, please clarify and organize – unless I am missing something. 

A10) Thanks for your comment. The requested modifies were made

Q11) Discussion:  Last sentence “Further studies . . . “.  Please delete.  You have addressed the limitations fo the study.

A11) Thanks for your comment. The requested modifies were made

Reviewer 2 Report

The authors present a systematic review seeking to describe and compare the use and outcomes of dynamic and rigid splints for the treatment of developmental dysplasia of the hip. 

The manuscript is clear, concise and well-written overall. The major point of concern is in the search strategy, utilizing only two databases to perform the search, and apparently using only a few generic keywords. More rigorous search term strategy including MESH terms and advanced logic to combine strategies would be likely more effective at identifying appropriate sources. Perhaps this strategy was applied, but should be more clearly described in the methods either via figure or description. I would consider refining the search and expanding to some of the commonly used databases for systematic reviews including medline, EMBASE, Ovid etc.

In terms of inclusion/exclusion criteria, was there a maximum age considered or all ages included and just noted?

Please note whether any studies were added from the manual reference check of included full-text review studies, and if so, how many. 

Table 2 - Heading "Mean age of the brace (weeks)" - is this mean age of the patient AT the time of brace application? This heading is unclear.

Would be beneficial to note the length of brace application (if reported) in the studies. This valuable information to know as brace length may be associated with outcomes and complications.  

Table 3 appears to be duplicated in the text.

Throughout the manuscript - Ifeld brace should be referring to the Ilfeld brace.

Author Response

The authors present a systematic review seeking to describe and compare the use and outcomes of dynamic and rigid splints for the treatment of developmental dysplasia of the hip.

The manuscript is clear, concise and well-written overall.

Q1) The major point of concern is in the search strategy, utilizing only two databases to perform the search, and apparently using only a few generic keywords. More rigorous search term strategy including MESH terms and advanced logic to combine strategies would be likely more effective at identifying appropriate sources. Perhaps this strategy was applied, but should be more clearly described in the methods either via figure or description. I would consider refining the search and expanding to some of the commonly used databases for systematic reviews including medline, EMBASE, Ovid etc.

A1) Thanks for your comment. The described keywords or MeSH terms were combined in order to achieve the maximum research efficacy. As you suggested the research strategy were clarified in the proper section.

Q2) In terms of inclusion/exclusion criteria, was there a maximum age considered or all ages included and just noted?

A2) Thanks for your comment. The criterium was clarified.

Q3) Please note whether any studies were added from the manual reference check of included full-text review studies, and if so, how many.

A3) Thanks for the comment. The requested clarification was added.

Q4) Table 2 - Heading "Mean age of the brace (weeks)" - is this mean age of the patient AT the time of brace application? This heading is unclear.

A4) Thanks for the comment. The requested heading was clarified

Q5) Would be beneficial to note the length of brace application (if reported) in the studies. This valuable information to know as brace length may be associated with outcomes and complications. 

A5) Thanks for the comment, we appreciate your suggestion; unfortunately, some authors did not report the mean follow-up time, or any brace application timing in their studies. For these reasons, the analysis could be influenced by a selection risk bias.

Q6) Table 3 appears to be duplicated in the text.

A6) Thank for you comment. The duplicated table was removed.

Q7) Throughout the manuscript - Ifeld brace should be referring to the Ilfeld brace.

A7) Thank for you comment. The typos were corrected.

Reviewer 3 Report

Table 1 - there are 18 papers - authors selected 19 - top up

Table 2 - there are 25 positions not 19 - some of the papers were listed more than once - a few are for different braces, explain in the text

position 28 is Wilkinson not Wllkinson (table 1 and 2, page 8 position 3.4)

page 6 under figure 1 "In these 19 studies, a total of 5100 patients and 6755 hips were identified. Overall, a success with splintage was observed in 6272 hips with a total success rate of 93%." the same sentence repeated on page 7 - first one should be delate

Author Response

Q1) Table 1 - there are 18 papers - authors selected 19 - top up

A1) Thanks for your suggestion. The missing authors was included

Q2) Table 2 - there are 25 positions not 19 - some of the papers were listed more than once - a few are for different braces, explain in the text

A2) Thanks for your suggestion. Explanation was added in results.

Q3) position 28 is Wilkinson not Wllkinson (table 1 and 2, page 8 position 3.4)

A3) Thanks for your suggestion. the typos were corrected.

Q4) page 6 under figure 1 "In these 19 studies, a total of 5100 patients and 6755 hips were identified. Overall, a success with splintage was observed in 6272 hips with a total success rate of 93%." the same sentence repeated on page 7 - first one should be delate

A4) Thanks for your suggestion. the phrase was re-written.

Round 2

Reviewer 2 Report

Thank you for effectively and efficiently addressing my concerns with this article.